# Identification and Overidentification of Linear Structural Equation Models

**Bryant Chen**
University of California, Los Angeles
Computer Science Department
Los Angeles, CA, 90095-1596, USA

## Abstract

In this paper, we address the problems of identifying linear structural equation models and discovering the constraints they imply. We first extend the half-trek criterion to cover a broader class of models and apply our extension to finding testable constraints implied by the model. We then show that any semi-Markovian linear model can be recursively decomposed into simpler sub-models, resulting in improved identification and constraint discovery power. Finally, we show that, unlike the existing methods developed for linear models, the resulting method subsumes the identification and constraint discovery algorithms for non-parametric models.

## 1 Introduction

Many researchers, particularly in economics, psychology, and the social sciences, use linear structural equation models (SEMs) to describe the causal and statistical relationships between a set of variables, predict the effects of interventions and policies, and to estimate parameters of interest. When modeling using linear SEMs, researchers typically specify the causal structure (i.e. exclusion restrictions and independence restrictions between error terms) from domain knowledge, leaving the structural coefficients (representing the strength of the causal relationships) as free parameters to be estimated from data. If these coefficients are known, then total effects, direct effects, and counterfactuals can be computed from them directly (Balke and Pearl, 1994). However, in some cases, the causal assumptions embedded in the model are not enough to uniquely determine one or more coefficients from the probability distribution, and therefore, cannot be estimated using data. In such cases, we say that the coefficient is *not identified* or *not identifiable*[1].

In other cases, a coefficient may be *overidentified* in addition to being identified, meaning that there are at least two minimal sets of logically independent assumptions in the model that are sufficient for identifying a coefficient, and the identified expressions for the coefficient are distinct functions of the covariance matrix (Pearl, 2004). As a result, the model imposes a testable constraint on the probability distribution that the two (or more) identified expressions for the coefficient are equal.

As compact and transparent representations of the model's structure, causal graphs provide a convenient tool to aid in the identification of coefficients. First utilized as a causal inference tool by Wright (1921), graphs have more recently been applied to identify causal effects in non-parametric causal models (Pearl, 2009) and enabled the development of causal effect identification algorithms that are complete for non-parametric models (Huang and Valtorta, 2006; Shpitser and Pearl, 2006). These algorithms can be applied to the identification of coefficients in linear SEMs by identifying non-parametric direct effects, which are closely related to structural coefficients (Tian, 2005; Chen and Pearl, 2014). Algorithms designed specifically for the identification of linear SEMs were de-

veloped by Brito and Pearl (2002), Brito (2004), Tian (2005, 2007, 2009), Foygel et al. (2012), and Chen et al. (2014).

Graphs have also proven to be valuable tools in the discovery of testable implications. It is well known that conditional independence relationships can be easily read from the causal graph using d-separation (Pearl, 2009), and Kang and Tian (2009) gave a procedure for linear SEMs that enumerates a set of conditional independences that imply all others. In non-parametric models without latent variables or correlated error terms, these conditional independence constraints represent all of the testable implications of the model (Pearl, 2009). In models with latent variables and/or correlated error terms, there may be additional constraints implied by the model. These non-independence constraints, often called *Verma constraints*, were first noted by Verma and Pearl (1990), and Tian and Pearl (2002b) and Shpitser and Pearl (2008) developed graphical algorithms for systematically discovering such constraints in non-parametric models. In the case of linear models, Chen et al. (2014) applied their aforementioned identification method to the discovery of overidentifying constraints, which in some cases are equivalent to the non-parametric constraints enumerated in Tian and Pearl (2002b) and Shpitser and Pearl (2008).

Surprisingly, naively applying algorithms designed for non-parametric models to linear models enables the identification of coefficients and constraints that the aforementioned methods developed for linear models are unable to, despite utilizing the additional assumption of linearity. In this paper, we first extend the half-trek identification method of Foygel et al. (2012) and apply it to the discovery of *half-trek constraints*, which generalize the overidentifying constraints given in Chen et al. (2014). Our extensions can be applied to Markovian, semi-Markovian, and non-Markovian models. We then demonstrate how recursive c-component decomposition, which was first utilized in identification algorithms developed for non-parametric models (Tian, 2002; Huang and Valtorta, 2006; Shpitser and Pearl, 2006), can be incorporated into our linear identification and constraint discovery methods for Markovian and semi-Markovian models. We show that doing so allows the identification of additional models and constraints. Further, we will demonstrate that, unlike existing algorithms, our method subsumes the aforementioned identification and constraint discovery methods developed for non-parametric models when applied to linear SEMs.

## 2 Preliminaries

A linear structural equation model consists of a set of equations of the form, $X = \Lambda X + \epsilon$, where $X = [x_1, ..., x_n]^t$ is a vector containing the model variables, $\Lambda$ is a matrix containing the *coefficients* of the model, which convey the strength of the causal relationships, and $\epsilon = [\epsilon_1, ..., \epsilon_n]^t$ is a vector of error terms, which represents omitted or latent variables. The matrix $\Lambda$ contains zeroes on the diagonal, and $\Lambda_{ij} = 0$ whenever $x_i$ is not a cause of $x_j$. The error terms are normally distributed random variables and induce the probability distribution over the model variables. The covariance matrix of $X$ will be denoted by $\Sigma$ and the covariance matrix over the error terms, $\epsilon$, by $\Omega$.

An instantiation of a model $M$ is an assignment of values to the model parameters (i.e. $\Lambda$ and the non-zero elements of $\Omega$). For a given instantiation $m_i$, let $\Sigma(m_i)$ denote the covariance matrix implied by the model and $\lambda_k(m_i)$ be the value of coefficient $\lambda_k$.

**Definition 1.** *A coefficient, $\lambda_k$, is* identified *if for any two instantiations of the model, $m_i$ and $m_j$, we have $\lambda_k(m_i) = \lambda_k(m_j)$ whenever $\Sigma(m_i) = \Sigma(m_j)$.*

In other words, $\lambda_k$ is identified if it can be uniquely determined from the covariance matrix, $\Sigma$. Now, we define when a structural coefficient, $\lambda_k$, is *overidentified*.

**Definition 2.** *(Pearl, 2004) A coefficient, $\lambda_k$ is* overidentified *if there are two or more distinct sets of logically independent assumptions in $M$ such that*

*(i) each set is* sufficient *for deriving $\lambda_k$ as a function of $\Sigma$, $\lambda_k = f(\Sigma)$,*

*(ii) each set induces a* distinct *function $\lambda_k = f(\Sigma)$, and*

*(iii) each assumption set is* minimal*, that is, no proper subset of those assumptions is sufficient for the derivation of $\lambda_k$.*

The causal graph or path diagram of an SEM is a graph, $G = (V, D, B)$, where $V$ are vertices or nodes, $D$ directed edges, and $B$ bidirected edges. The vertices represent model variables. Directed

eges represent the direction of causality, and for each coefficient $\Lambda_{ij} \neq 0$, an edge is drawn from $x_i$ to $x_j$. Each directed edge, therefore, is associated with a coefficient in the SEM, which we will often refer to as its structural coefficient. The error terms, $\epsilon_i$, are not represented in the graph. However, a bidirected edge between two variables indicates that their corresponding error terms may be statistically dependent while the lack of a bidirected edge indicates that the error terms are independent. When the causal graph is acyclic without bidirected edges, then we say that the model is *Markovian*. Graphs with bidirected edges are *non-Markovian*, while acyclic graphs with bidirected edges are additionally called *semi-Markovian*.

We will use standard graph terminology with $Pa(y)$ denoting the parents of $y$, $Anc(y)$ denoting the ancestors of $y$, $De(y)$ denoting the descendants of $y$, and $Sib(y)$ denoting the siblings of $y$, the variables that are connected to $y$ via a bidirected edge. $He(E)$ denotes the heads of a set of directed edges, $E$, while $Ta(E)$ denotes the tails. Additionally, for a node $v$, the set of edges for which $He(E) = v$ is denoted $Inc(v)$. Lastly, we will utilize d-separation (Pearl, 2009).

Lastly, we establish a couple preliminary definitions around half-treks. These definitions and illustrative examples can also be found in Foygel et al. (2012) and Chen et al. (2014).

**Definition 3.** *(Foygel et al., 2012) A* half-trek*, $\pi$, from $x$ to $y$ is a path from $x$ to $y$ that either begins with a bidirected arc and then continues with directed edges towards $y$ or is simply a directed path from $x$ to $y$.*

We will denote the set of nodes that are reachable by half-trek from $v$ $htr(v)$.

**Definition 4.** *(Foygel et al., 2012) For any half-trek, $\pi$, let* Right$(\pi)$ *be the set of vertices in $\pi$ that have an outgoing directed edge in $\pi$ (as opposed to bidirected edge) union the last node in the trek. In other words, if the trek is a directed path then every node in the path is a member of Right($\pi$). If the trek begins with a bidirected edge then every node other than the first node is a member of Right($\pi$).*

**Definition 5.** *(Foygel et al., 2012) A system of half-treks, $\pi_1, ..., \pi_n$, has* no sided intersection *if for all $\pi_i, \pi_j \in \{\pi_1, ..., \pi_n\}$ such that $\pi_i \neq \pi_j$, Right($\pi_i$)∩Right($\pi_j$)= $\emptyset$.*

**Definition 6.** *(Chen et al., 2014) For an arbitrary variable, $v$, let $Pa_1, Pa_2, ..., Pa_k$ be the unique partition of Pa(v) such that any two parents are placed in the same subset, $Pa_i$, whenever they are connected by an unblocked path (given the empty set). A* connected edge set *with head $v$ is a set of directed edges from $Pa_i$ to $v$ for some $i \in \{1, 2, ..., k\}$.*

# 3   General Half-Trek Criterion

The half-trek criterion is a graphical condition that can be used to determine the identifiability of recursive and non-recursive linear models (Foygel et al., 2012). Foygel et al. (2012) use the half-trek criterion to identify the model variables one at a time, where each identified variable may be able to aid in the identification of other variables. If any variable is not identifiable using the half-trek criterion, then their algorithm returns that the model is not *HTC-identifiable*. Otherwise the algorithm returns that the model is identifiable. Their algorithm subsumes the earlier methods of Brito and Pearl (2002) and Brito (2004). In this section, we extend the half-trek criterion to allow the identification of arbitrary subsets of edges belonging to a variable. As a result, our algorithm can be utilized to identify as many coefficients as possible, even when the model is not identified. Additionally, this extension improves our ability to identify entire models, as we will show.

**Definition 7.** *(General Half-Trek Criterion) Let $E$ be a set of directed edges sharing a single head $y$. A set of variables $Z$ satisfies the* general half-trek criterion *with respect to $E$, if*

*(i)  $|Z| = |E|$,*

*(ii)  $Z \cap (y \cup Sib(y)) = \emptyset$,*

*(iii)  There is a system of half-treks with no sided intersection from $Z$ to $Ta(E)$, and*

*(iv)  $(Pa(y) \setminus Ta(E)) \cap htr(Z) = \emptyset$.*

A set of directed edges, $E$, sharing a head $y$ is identifiable if there exists a set, $Z_E$, that satisfies the general half-trek criterion (g-HTC) with respect to $E$, and $Z_E$ consists only of "allowed" nodes. Intuitively, a node $z$ is allowed if $E_{zy}$ is identified or empty, where $E_{zy} \subseteq Inc(z)$ is the set of edges

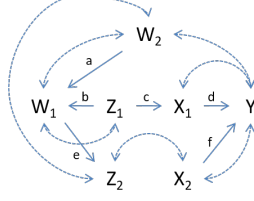

Figure 1: The above model is identified using the g-HTC but not the HTC.

belonging to $z$ that lie on half-treks from $y$ to $z$ or lie on unblocked paths (given the empty set) between $z$ and $Pa(y) \setminus Ta(E)$.[2] The following definition formalizes this notion.

**Definition 8.** *A node, $z$, is g-HT allowed (or simply* allowed*) for directed edges $E$ with head $y$ if $E_{zy} = \emptyset$ or there exists sequences of sets of nodes, $(Z_1, ...Z_k)$, and sets of edges, $(E_1, ..., E_k)$, with $E_{zy} \subseteq E_1 \cup ... \cup E_k$ such that*

 *(i) $Z_i$ satisfies the g-HTC with respect to $E_i$ for all $i \in \{1, ..., k\}$,*

 *(ii) $E_{Z_1 y_1} = \emptyset$, where $y_i = He(E_i)$ for all $i \in \{1, ..., k\}$, and*

 *(iii) $E_{Z_i y_i} \subseteq (E_1 \cup ... \cup E_{i-1})$ for all $i \in \{1, ...k\}$.*

When a set of allowed nodes, $Z_E$, satisfies the g-HTC for a set of edges $E$, then we will say that $Z_E$ is a *g-HT admissible* set for $E$.

**Theorem 1.** *If a g-HT admissible set for directed edges $E_y$ with head $y$ exists then $E_y$ is g-HT identifiable. Further, let $Z_{E_y} = \{z_1, ..., z_k\}$ be a g-HT admissible set for $E_y$, $Ta(E_y) = \{p_1, ..., p_k\}$, and $\Sigma$ be the covariance matrix of the model variables. Define $\mathbf{A}$ as*

$$\mathbf{A_{ij}} = \begin{cases} [(I - \Lambda)^T \Sigma]_{z_i p_j}, & E_{z_i y} \neq \emptyset \\ \Sigma_{z_i p_j}, & E_{z_i y} = \emptyset \end{cases} \tag{1}$$

*and $\mathbf{b}$ as*

$$\mathbf{b_i} = \begin{cases} [(I - \Lambda)^T \Sigma]_{z_i y}, & E_{z_i y} \neq \emptyset \\ \Sigma_{z_i y}, & E_{z_i y} = \emptyset \end{cases} \tag{2}$$

*Then $\mathbf{A}$ is an invertible matrix and $\mathbf{A} \cdot \Lambda_{Ta(E_y), y} = \mathbf{b}$.*

*Proof.* See Appendix for proofs of all theorems and lemmas. □

The g-HTC impoves upon the HTC because subsets of a variable's coefficients may be identifiable even when the variable is not. By identifying subsets of a variable's coefficients, we not only allow the identification of as many coefficients as possible in unidentified models, but we also are able to identify additional models as a whole. For example, Figure 1 is not identifiable using the HTC. In order to identify $Y$, $Z_2$ needs to be identified first as it is the only variable with a half-trek to $X_2$ without being a sibling of $Y$. However, to identify $Z_2$, either $Y$ or $W_1$ needs to be identified. Finally to identify $W_1$, $Y$ needs to be identified. This cycle implies that the model is not HTC-identifiable. It is, however, g-HTC identifiable since the g-HTC allows $d$ to be identified independently of $f$, using $\{Z_1\}$ as a g-HT admissible set, which in turn allows $\{Y\}$ to be a g-HT admissible set for $W_1$'s coefficient, $a$.

Finding a g-HT admissible set for directed edges, $E$, with head, $y$, from a set of allowed nodes, $A_E$, can be accomplished by utilizing the max-flow algorithm described in Chen et al. (2014)[3], which we call $\mathrm{MaxFlow}(G, E, A_E)$. This algorithm returns a maximal set of allowed nodes that satisfies (ii) - (iv) of the g-HTC.

In some cases, there may be no g-HT admissible set for $E'$ but there may be one for $E \subset E'$. In other cases, there may be no g-HT admissible set of variables for a set of edges $E$ but there may be a

[2] We will continue to use the $E_{Zy}$ notation and allow $Z$ to be a set of nodes.
[3] Brito (2004) utilized a similar max-flow construction in his identification algorithm.

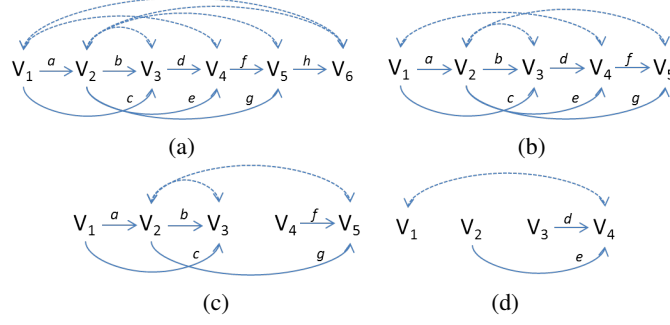

Figure 2: (a) The graph is not identified using the g-HTC and cannot be decomposed (b) After removing $V_6$ we are able to decompose the graph (c) Graph for c-component, $\{V_2, V_3, V_5\}$ (d) Graph for c-component, $\{V_1, V_4\}$

g-HT admissible set of variables for $E'$ with $E \subset E'$. As a result, if a HT-admissible set does not exist for $E_y$, where $E_y = Inc(y)$ for some node $y$, we may have to check whether such a set exists for all possible subsets of $E_y$ in order to identify as many coefficients in $E_y$ as possible. This process can be somewhat simplified by noting that if $E$ is a connected edge set with no g-HT admissible set, then there is no superset $E'$ with a g-HT admissible set.

An algorithm that utilizes the g-HTC and Theorem 1 to identify as many coefficients as possible in recursive or non-recursive linear SEMs is given in the Appendix. Since we may need to check the identifiability of all subsets of a node's edges, the algorithm's complexity is polynomial time if the degree of each node is bounded.

## 4 Generalizing Overidentifying Constraints

Chen et al. (2014) discovered overidentifying constraints by finding two HT-admissible sets for a given connected edge set. When two such sets exist, we obtain two distinct expressions for the identified coefficients, and equating the two expressions gives the overidentifying constraint. However, we may be able to obtain constraints even when $|Z_E| < |E|$ and $E$ is not identified. The algorithm, MaxFlow, returns a maximal set, $Z_E$, for which the equations, $\mathbf{A} \cdot \Lambda_{Ta(E),y} = \mathbf{b}$, are linearly independent, regardless of whether $|Z_E| = |E|$ and $E$ is identified or not. Therefore, if we are able to find an allowed node $w$ that satisfies the conditions below, then the equation $a_w \cdot \Lambda_{Ta(E),y} = b_w$ will be a linear combination of the equations, $\mathbf{A} \cdot \Lambda_{Ta(E),y} = \mathbf{b}$.

**Theorem 2.** *Let $Z_E$ be a set of maximal size that satisfies conditions (ii)-(iv) of the g-HTC for a set of edges, $E$, with head $y$. If there exists a node $w$ such that there exists a half-trek from $w$ to Ta(E), $w \notin (y \cup Sib(y))$, and $w$ is g-HT allowed for $E$, then we obtain the equality constraint, $\mathbf{a_w} \mathbf{A}_{\mathrm{right}}^{-1} \mathbf{b} = b_w$, where $\mathbf{A}_{\mathrm{right}}^{-1}$ is the right inverse of $\mathbf{A}$.*

We will call these generalized overidentifying constraints, *half-trek constraints* or *HT-constraints*. An algorithm that identifies coefficients and finds HT-constraints for a recursive or non-recursive linear SEM is given in the Appendix.

## 5 Decomposition

Tian showed that the identification problem could be simplified in semi-Markovian linear structural equation models by decomposing the model into sub-models according to their *c-components* (Tian, 2005). Each coefficient is identifiable if and only if it is identifiable in the sub-model to which it belongs (Tian, 2005). In this section, we show that the c-component decomposition can be applied recursively to the model after marginalizing certain variables. This idea was first used to identify interventional distributions in non-parameteric models by Tian (2002) and Tian and Pearl (2002a) and adapting this technique for linear models will allow us to identify models that the g-HTC, even coupled with (non-recursive) c-component decomposition, is unable to identify. Further, it ensures the identification of all coefficients identifiable using methods developed for non-parametric models–a guarantee that none of the existing methods developed for linear models satisfy.

The graph in Figure 2a consists of a single c-component, and we are unable to decompose it. As a result, we are able to identify $a$ but no other coefficients using the g-HTC. Moreover, $f = \frac{\partial}{\partial v_4} E[v_5|do(v_6, v_4, v_3, v_2, v_1)]$ is identified using identification methods developed for non-parametric models (e.g. do-calculus) but not the g-HTC or other methods developed for linear models.

However, if we remove $v_6$ from the analysis, then the resulting model can be decomposed. Let $M$ be the model depicted in Figure 2a, $P(v)$ be the distribution induced by $M$, and $M'$ be a model that is identical to $M$ except the equation for $v_6$ is removed. $M'$ induces the distribution $\int_{v_6} P(V) dv_6$, and its associated graph $G'$ yields two c-components, as shown in Figure 2b.

Now, decomposing $G'$ according to these c-components yields the sub-models depicted by Figures 2c and 2d. Both of these sub-models are identifiable using the half-trek criterion. Thus, all coefficients other than $h$ have been shown to be identifiable. Returning to the graph prior to removal, depicted in Figure 2a, we are now able to identify $h$ because both $v_4$ and $v_5$ are now allowed nodes for $h$, and the model is identified[4].

As a result, we can improve our identification and constraint-discovery algorithm by recursively decomposing, using the g-HTC and Theorem 2, and removing descendant sets[5]. Note, however, that we must consider every descendant set for removal. It is possible that removing $D_1$ will allow identification of a coefficient but removing a superset $D_2$ with $D_1 \subset D_2$ will not. Additionally, it is possible that removing $D_2$ will allow identification but removing a subset $D_1$ will not.

After recursively decomposing the graph, if some of the removed variables were unidentified, we may be able to identify them by returning to the original graph prior to removal since we may have a larger set of allowed nodes. For example, we were able to identify $h$ in Figure 2a by "un-removing" $v_6$ after the other coefficients were identified. In some cases, however, we may need to again recursively decompose and remove descendant sets. As a result, in order to fully exploit the powers of decomposition and the g-HTC, we must repeat the recursive decomposition process on the original model until all marginalized nodes are identified or no new coefficients are identified in an iteration.

Clearly, recursive decomposition also aids in the discovery of HT-constraints in the same way that it aids in the identification of coefficients using the g-HTC. However, note that recursive decomposition may also introduce additional d-separation constraints. Prior to decomposition, if a node $Z$ is d-separated from a node $V$ then we trivially obtain the constraint that $\Sigma_{ZV} = 0$. However, in some cases, $Z$ may become d-separated from $V$ after decomposition. In this case, the independence constraint on the covariance matrix of the decomposed c-component corresponds to a non-conditional independence constraint in the original joint distribution $P(V)$. It is for this reason that we output independence constraints in Algorithm 2 (see Appendix).

For example, consider the graph depicted in Figure 3a. Theorem 2 does not yield any constraints for the edges of $V_7$. However, after decomposing the graph we obtain the c-component for $\{V_2, V_5, V_7\}$, shown in Figure 3b. In this graph, $V_1$ is d-separated from $V_7$ yielding a non-independence constraint in the original model.

We can systematically identify coefficients and HT-constraints using recursive c-component decomposition by repeating the following steps for the model's graph $G$ until the model has been identified or no new coefficients are identified in an iteration:

(i) Decompose the graph into c-components, $\{S_i\}$

(ii) For each c-component, utilize the g-HTC and Theorems 1 and 2 to identify coefficients and find HT-constraints

(iii) For each descendant set, marginalize the descendant set and repeat steps (i)-(iii) until all variables have been marginalized

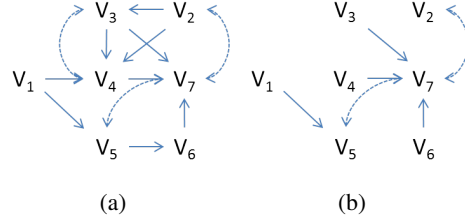

Figure 3: (a) $V_1$ cannot be d-separated from $V_7$ (b) $V_1$ is d-separated from $V_7$ in the graph of the c-component, $\{V_2, V_5, V_7\}$

If a coefficient $\alpha$ can be identified using the above method (see also Algorithm 3 in the Appendix, which utilizes recursive decomposition to identify coefficients and output HT-constraints), then we will say that $\alpha$ is *g-HTC identifiable*.

We now show that any direct effect identifiable using non-parametric methods is also g-HTC identifiable.

**Theorem 3.** *Let $M$ be a linear SEM with variables $V$. Let $M'$ be a non-parametric SEM with identical structure to $M$. If the direct effect of $x$ on $y$ for $x, y \in V$ is identified in $M'$ then the coefficient $\Lambda_{xy}$ in $M$ is g-HTC identifiable and can be identified using Algorithm 3 (see Appendix).*

## 6 Non-Parametric Verma Constraints

Tian and Pearl (2002b) and Shpitser and Pearl (2008) provided algorithms for discovering Verma constraints in recursive, non-parametric models. In this section, we will show that the constraints obtained by the above method and Algorithm 3 (see Appendix) subsume the constraints discovered by both methods when applied to linear models. First, we will show that the constraints identified in (Tian and Pearl, 2002b), which we call *Q-constraints*, are subsumed by HT-constraints. Second, we will show that the constraints given by Shpitser and Pearl (2008), called dormant independences, are, in fact, equivalent to the constraints given by Tian and Pearl (2002b) for linear models. As a result, both dormant independences and Q-constraints are subsumed by HT-constraints.

### 6.1 Q-Constraints

We refer to the constraints enumerated in (Tian and Pearl, 2002b) as Q-constraints since they are discovered by identifying Q-factors, which are defined below.

**Definition 9.** *For any subset, $S \subseteq V$, the Q-factor, $Q_S$, is given by*

$$Q_S = \int_{\epsilon_S} \prod_{i | V_i \in S} P(v_i | pa_i, \epsilon_i) P(\epsilon_S) d\epsilon_S, \tag{3}$$

*where $\epsilon_S$ contains the error terms of the variables in $S$.*

A Q-factor, $Q_S$, is identifiable whenever $S$ is a c-component (Tian and Pearl, 2002a).

**Lemma 1.** *(Tian and Pearl, 2002a) Let $\{v_1, ..., v_n\}$ be sorted topologically, $S$ be a c-component, $V^{(i)} = \{v_1, ..., v_i\}$, and $V^{(0)} = \emptyset$. Then $Q_S$ can be computed as $Q_S = \prod_{\{i | v_i \in S\}} P(v_i | V^{(i-1)}), j = 1, ..., k$.*

For example, consider again Figure 2b. We have that $Q_1 = P(v_1)P(v_4|v_3, v_2, v_1)$ and $Q_2 = P(v_2|v_1)P(v_3|v_2, v_1)P(v_5|v_4, v_3, v_2, v_1)$.

A Q-factor can also be identified by marginalizing out descendant sets (Tian and Pearl, 2002a). Suppose that $Q_S$ is identified and $D$ is a descendant set in $G_S$, then

$$Q_{S_i \setminus D} = \sum_D Q_{S_i}. \tag{4}$$

If the marginalization over $D$ yields additional c-components in the marginalized graph, then we can again compute each of them from $Q_{S \setminus D}$ (Tian and Pearl, 2002b).

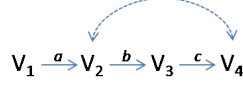

Figure 4: The above graph induces the Verma constraint, $Q[v_4]$ is not a function of $v_1$, and equivalently, $v_4 \perp v_1 | do(v_3)$.

Tian's method recursively computes the Q-factors associated with c-components, marginalizes descendant sets in the graph for the computed Q-factor, and again computes Q-factors associated with c-components in the marginalized graph. The Q-constraint is obtained in the following way. The definition of a Q-factor, $Q_S$, given by Equation 3 is a function of $Pa(S)$ only. However, the equivalent expression given by Lemma 1 and Equation 4 may be functions of additional variables.

For example, in Figure 4, $\{v_2, v_4\}$ is a c-component so we can identify $Q_{v_2 v_4} = P(v_4 | v_3, v_2, v_1)P(v_2 | v_1)$. The decomposition also makes $v_2$ a leaf node in $G_{v_2 v_4}$. As a result, we can identify $Q_{v_4} = \int_{v_2} P(v_4 | v_3, v_2, v_1)P(v_2 | v_1)dv_2$. Since $v_1$ is not a parent of $v_4$ in $G_{v_4}$, we have that $Q_{v_4} = \int_{v_2} P(v_4 | v_3, v_2, v_1)P(v_2 | v_1)dv_2 \perp v_1$.

**Theorem 4.** *Any Q-constraint, $Q_S \perp Z$, in a linear SEM, has an equivalent set of HT-constraints that can be discovered using Algorithm 3 (see Appendix).*

## 6.2 Dormant Independences

Dormant independences have a natural interpretation as independence and conditional independence constraints within identifiable interventional distributions (Shpitser and Pearl, 2008). For example, in Figure 4, the distribution after intervention on $v_3$ can be represented graphically by removing the edge from $v_2$ to $v_3$ since $v_3$ is no longer a function of $v_2$ but is instead a constant. In the resulting graph, $v_4$ is d-separated from $v_1$ implying that $v_4$ is independent of $v_1$ in the distribution, $P(v_4, v_2, v_1 | do(v_3))$. In other words, $P(v_4 | do(v_3), v_1) = P(v_4 | do(v_3))$. Now, it is not hard to show that $P(v_4 | v_1, do(v_3))$ is identifiable and equal to $\sum_{v_2} P(v_4 | v_3, v_2, v_1)P(v_2 | v_1)$ and we obtain the constraint that $\sum_{v_2} P(v_4 | v_3, v_2, v_1)P(v_2 | v_1)$ is not a function of $v_1$, which is exactly the Q-constraint we obtained above.

It turns out that dormant independences among singletons and Q-constraints are equivalent, as stated by the following lemma.

**Lemma 2.** *Any dormant independence, $x \perp\!\!\!\perp y | w, do(Z)$, with $x$ and $y$ singletons has an equivalent Q-constraint and vice versa.*

Since pairwise independence implies independence in normal distributions, Lemma 2 and Theorem 4 imply the following theorem.

**Theorem 5.** *Any dormant independence among sets, $x \perp\!\!\!\perp y | W, do(Z)$, in a linear SEM, has an equivalent set of HT-constraints that can be discovered by incorporating recursive c-component decomposition with Algorithm 3 (see Appendix).*

## 7 Conclusion

In this paper, we extend the half-trek criterion (Foygel et al., 2012) and generalize the notion of overidentification to discover constraints using the generalized half-trek criterion, even when the coefficients are not identified. We then incorporate recursive c-component decomposition and show that the resulting identification method is able to identify more models and constraints than the existing linear and non-parameteric algorithms.

Finally, we note that while we were preparing this manuscript for submission, Drton and Weihs (2016) independently introduced a similar idea to the recursive decomposition discussed in this paper, which they called ancestor decomposition. While ancestor decomposition is more efficient, recursive decomposition is more general in that it enables the identification of a larger set of coefficients.

# 8  Acknowledgments

I would like to thank Jin Tian and Judea Pearl for helpful comments and discussions. This research was supported in parts by grants from NSF #IIS-1302448 and #IIS-1527490 and ONR #N00014-13-1-0153 and #N00014-13-1-0153.

## Footnotes

[1]We will also use the term "identified" with respect to individual variables and the model as a whole.

[4]While $v_4$ and $v_5$ are technically not allowed according to Definition 8, they can be used in g-HT admissible sets to identify $h$ using Theorem 1 since their coefficients have been identified.

[5]Only removing descendant sets have the ability to break up components. For example, removing $\{v_2\}$ from Figure 2a does not break the c-component because removing $v_2$ would relegate its influence to the error term of its child, $v_3$. As a result, the graph of the resulting model would include a bidirected arc between $v_3$ and $v_6$, and we would still have a single c-component.

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
