[Reviews · NeurIPS 2016]

Reviewer 1

Summary

The paper proposes methods for identification of linear structural equation models. This is done by first extending the half-trek criterion, and recursively decomposing SEMs into smaller submodels and using the criterion there. The new method subsumes identification power for non-parameteric causal effect identification. The results are interesting, but their full potential is not really explained or demonstrated. The text and presentation could be improved.

Qualitative Assessment

Technical quality: -no simulations or implementations presented, the added identification is only given by a few examples Novelty: -some new coefficients are identified and testable constraints produced -contribution seems incremental to already established results - or at least it is presented so Potential Impact: -the results build for complete identification and added testable constraints -The conclusion does not explain potential impact, a discussion of this would help a lot Clarity and presentation: -the text presents a huge number of definitions and theorems. It is on the level of a technical report for the writers themselves. It is impossible for the reader to keep the definitions in mind when continuing to read the paper. The text should be developed more into direction of clearly explaining _why_ and _how_ and simplified. -edge heads are top of each other in graphs, the graphs are ambiguous, this should be fixed Before definition 1: You claim that one can standardize the model variables without losing generality, I tend to believe this, but where is it shown? You should cite it here explicitly. ----- After author feedback updated scores in clarity and technical quality from 2 to 3.

Confidence in this Review

1-Less confident (might not have understood significant parts)


Reviewer 2

Summary

The paper extends the half-trek criterion to cover both semi-Markovian and non-Markovian models. The work also incorporates recursive c-component decomposition to show the resulting identification method is able to identify more models and constraints than existing algorithms.

Qualitative Assessment

The identification problem of the Linear Structural Equation Models is fundamental and important to the SEM research. This work extend the half-trek criterion to cover a broader class of models, and it is an important theoretical advancement of this area.

Confidence in this Review

2-Confident (read it all; understood it all reasonably well)


Reviewer 3

Summary

This paper proposes a method of identifying linear structural equation models (SEMs) and thus discovering constraints inherent in the models. Specifically, the authors provided several definitions by extending the half-trek criterion to determine the identifiability of arbitrary subsets of edges belonging to a variable, thus to identify model variables. It is also shown that the c-component decomposition can be applied to the model after marginalizing certain variables recursively.

Qualitative Assessment

- Overall, the paper is written well with good definitions and examples. However, this reviewer is wondering how well the proposed method is working for real applications. While the proposed method sounds reasonable enough, it is highly recommended to provide the experimental results on synthetic and/or real data. In such a way, the potential readers could better understand how to use the proposed method for their own researches or practical applications. - Every criterion other than (iv) in Definition 7 were the same as the Edge Set Half-Trek in (Chen et al., 2004). It is noteworthy to describe or explain the role of the new criterion (iv) for “generalization”. * Minor issues - References in 337-338 and 341-342 are the same. - Page 3, line 94: for consistency, Y should be y. - Definition 6: Again for consistency, V should be v.

Confidence in this Review

1-Less confident (might not have understood significant parts)


Reviewer 4

Summary

The paper regards the identification of constraints in linear structural equation models (SEMs). It builds on existing work on half-trek criterion and recursive c-component decomposition.

Qualitative Assessment

The paper is well-written and as far as I can see correct.

Confidence in this Review

1-Less confident (might not have understood significant parts)


Reviewer 5

Summary

This paper establishes a generalization of the half-trek criterion (HTC) for identifying coefficients in linear structural equation models (LSEMs). The generalized criterion (g-HTC) is designed to identify the coefficients associated with a subset of edges into a variable when the set of all edges into that variable is not identified by the original HTC. It turns out, moreover, that there are models that are identifiable by the g-HTC but not by the HTC. In connection to the g-HTC, this paper also proposes a way to derive constraints implied by a linear structural equation model. Furthermore, it is shown that when combined with decomposition of semi-Markovian models into sub-models over its c-components and marginalization of descendant sets, the proposed method of identification can identify every coefficient in a semi-Markovian LSEM that is identifiable according to non-parametric criteria (such as the do-calculus), and the method of deriving constraints subsumes some non-parametric methods in the literature.

Qualitative Assessment

This is a solid contribution to the literature on the identification of linear models. The paper is well organized and for the most part well written. Due to limited space, the technical details (even including the supplementary material) are unfortunately not fully self-contained and not easy to follow for non-experts. It would have been more readable if more background (such as the original HTC and more information on the proof of the theorem on HTC-identifiability) had been included in the supplementary material. Still, most proofs look credible (though I did not have time to check all of them carefully; sorry about that.) I am puzzled, however, by the last part of the proof of Theorem 4; I would appreciate some elaboration of how HT constraints express the conditional independence constraints in question.

Confidence in this Review

2-Confident (read it all; understood it all reasonably well)